# FPGA Implementation of Mutual Authentication Protocol for Medication Security System

**Wei-Chen Lin [1], Po-Kai Huang [2], Chung-Long Pan [2] and Yu-Jung Huang [3],***

[1]  Medical Research Department, E-DA Hospital, Kaohsiung 82445, Taiwan; ed109415@edah.org.tw
[2]  Department of Electrical Engineering, I-Shou University, Kaohsiung 84001, Taiwan;
    b87401074@ntu.edu.tw (P.-K.H.); ptl@isu.edu.tw (C.-L.P.)
[3]  Department of Electronic Engineering, I-Shou University, Kaohsiung 84001, Taiwan
*  Correspondence: yjhuang@isu.edu.tw

**Abstract:** Medication safety administration is a complicated process involving the information of patients, drugs, and data storage. The sensitive data transmitted through wireless sensor networks (WSNs) from Internet of things (IoT) over an insecure channel is vulnerable to several threats and needs proper attention to be secured from adversaries. Taking medication safety into consideration, this paper presents a secure authentication protocol for wireless medical sensor networks. The XOR scheme-based algorithm is applied to achieve the purposes of data confidentiality. The proposed architecture is realized as hardware in a field-programmable gate array (FPGA) device which acts as a secure edge computing device. The performance of the proposed protocol is evaluated and simulated via Verilog hardware description language. The functionality of the proposed protocol is verified using the Altera Quartus II software tool and implemented in the Altera Cyclone II DE2-70 FPGA development module. Furthermore, the output signals from the FPGA are measured in the 16702A logic analyzer system to demonstrate real-time functional verification.

**Keywords:** field-programmable gate array; wireless medical sensor networks; mutual authentication protocol

## 1. Introduction

The Internet of things (IoT) as an emerging network paradigm has been broadly applied in our daily life along with the development of wireless sensor network (WSN) technologies [1,2]. WSNs have been subject to a great deal of research over recent years, driven by their potential to embed into the physical environment and observe and act on certain phenomena. Distinct characteristics arising from strict energy constraints, large numbers of nodes, and highly asymmetric data flows pose technical challenges for large-scale and long-term deployments. In 1978, the Defense Advanced Research Projects Agency (DARPA) was the first pioneer that organized the Distributed Sensor Nets Workshop, focusing on wireless sensor networks research challenges such as networking technologies, signal processing techniques, and distributed algorithms [3]. This led to the beginning of the worldwide research boom on WSNs technology. The integration of WSN applications and low-power sensing nodes with the Internet has been accomplished with various approaches and strategies [4,5].

WSNs have an autonomous and ubiquitous nature; the use of wireless sensor networks has emerged in numerous domains such as defense, agriculture, transportation, and healthcare. Wireless sensor networks for IoT-based healthcare systems have facilitated the next generation of the connected healthcare industry, thereby causing drift from the traditional approaches to revolutionized technological approaches [6]. The data collected in most applications are valuable and need to maintain security. However, it becomes very challenging to implement user authentication in WSN applications because of the limited resources available in sensor nodes.

The security protocols proposed for traditional WSNs are not directly appropriate for wireless medical sensor networks [7]. The main security concerns in medication security system architectures are to guarantee the authenticity and integrity of the commands issued by the medical requirement through the network system. Moreover, protocols that use complicated and computationally expensive cryptography may exhaust system resources (data/energy storage and communication range/bandwidth) [8,9]. For instance, processing and storing the cryptography data complicates the system design. The medication security system has unique and challenging operational and security requirements, particularly the development of lightweight authentication and low computational power. Field-programmable gate arrays (FPGAs) have recently become extremely popular due to their rapid prototyping, easy debugging, and reconfigurability characteristics. FPGAs have lower latency, low power consumption, and low-cost advantages, which makes them suitable for real-time applications such as automotive, health, and industrial applications. FPGAs are also well-suited to low-cost applications in which security and reliability are critical. In this work, a user authentication scheme is proposed and an FPGA is implemented to suit the infrastructure of a healthcare information system.

The remainder of the paper is organized as follows: Section 2 provides a background review; Section 3 explains the authentication scheme of the system; Section 4 shows the design and implementation results; Section 5 concludes the paper.

## 2. Background Review

### 2.1. Medication Errors

For a healthcare information system, medication management is a complex, multi-faceted operation involving multiple people and numerous data transmission steps; medication errors can occur throughout the system network. Medication error is an important cause of patient morbidity and mortality [10]. In efforts to prevent medication errors, the healthcare information system needs to give priority to keeping their networks secure by permitting only authenticated users or messages to gain access to their protected resources.

Nevertheless, it is challenging to provide consistent solutions to eliminate or minimize recurrent events and work toward improving patient safety [11]. If a proper medication system is proposed considering the circumstances of administrating medication safety, it will be able to help to prevent errors from occurring, whether it be a fault due to personnel or system failure. Due to advances in medical sensors and low-power network systems, the use of the IoT in electronic health (e-health) management systems has emerged in recent years to prevent the growing incidence of medical errors [12].

### 2.2. WSN Security and Privacy Issues

Because health information has always been regarded as highly sensitive personal data, privacy rules have been established to protect health information in many countries such as the Health Insurance Portability and Accountability Act (HIPAA) in the United States or Personal Data Protection Act (PDPA) in Taiwan [13,14]. However, technological advances have enabled healthcare information to be transmitted more effectively and provided more efficiently for the patients' healthcare. Recent advancements in wireless medical sensor networks are emerging as a significant component of next-generation healthcare systems. However, wireless communication networks are vulnerable to attacks, causing the receiver to receive the wrong information and causing unexpected damage. Protecting data confidentiality and privacy has become a very important issue [15]. Therefore, to ensure safe data transmission and access to information during transmission, proper security authentication is necessary to ensure that data can be correctly received at the destination. To reach this target, technologies such as encryption protect data privacy and integrity through a secure routing protocol to establish a secure route. On the use of wireless sensor networks, there must be common practice for each end party to send and receive keys and for a key management protocol to be used in pairs between the sensor node keys. Numerous studies have been proposed in the literature to provide cryptographic protec-

tion for tiny devices in constrained environments such as WSN based on the symmetric, public key, or hybrid encryption mechanisms [16,17]. In addition, many mechanisms have been proposed to improve the security of WSNs, such as intrusion detection and key management [18]. Since it provides basic security services by verifying the validity of user who wants to access the sensory data, identity authentication is also a crucial security mechanism for WSN [19].

## 3. Medication Safety WSN Authentication Scheme

Radiofrequency identification (RFID) and wireless sensors networks (WSNs) are two fundamental pillars that enable the Internet of things (IoT). Similar to radiofrequency identification, sensor networks have limited transmission capacity. A wireless medical sensor network (WMSN) can be used to construct a pervasive medical system [20]. However, the physical security of sensor nodes cannot be guaranteed because nodes are easily compromised by adversaries. For communication security and personal safety, it is necessary to realize mutual authentication on the Internet of sensors. To achieve local password change and forward security while resisting mobile device loss attacks, we propose a mutual authentication scheme for WMSN.

### 3.1. Mutual Authentication Algorithm

Following our previous study [21], the Padgen algorithm was proposed as the basis for mutual authentication, with the resulting key having the exclusive operation to generate a cover code and avoid vulnerabilities during the wireless transmission process. Wireless data transmission in a WMSN can be described as follows: if the node receives the correct password PW, then the user can access the message Msg. We assume that the Msg and PW are 32 bit message data and can be expressed as

$$Msg = a_0 \, a_1 \, a_2 \ldots a_{31}, \tag{1}$$

$$PW = P_0 \, P_1 \, P_2 \ldots P_{31}. \tag{2}$$

The random number is assumed to be 16 bit for the $R_t$ and $R_m$, expressed as follows:

$$R_t = d_{t1} \, d_{t2} \, d_{t3} \, d_{t4} \text{ (Base 16)}, \tag{3}$$

$$R_m = d_{m1} \, d_{m2} \, d_{m3} \, d_{m4} \text{ (Base 16)}. \tag{4}$$

The PAD can be generated using Msg and PW through the XOR-PadGen calculation method. The details of generating the key are expressed as follows:

$$\begin{aligned} \text{Msg-PadGen} \, (R_t, R_m) &= m_{dt1} \, m_{dt1+16} \, m_{dt2} \, m_{dt2+16} \| m_{dm1} \, m_{dm1+16} \, m_{dm2} \, m_{dm2+16} \\ &\quad \| m_{dt3} \, m_{dt3+16} \, m_{dm3} \, m_{dm3+16} \| m_{dt4} \, m_{dt4+16} \, m_{dm4} \, m_{dm4+16} \text{ (Base 2)}, \\ &= d_{v1} \, d_{v2} \, d_{v3} \, d_{v4} \text{ (Base 16)}. \end{aligned} \tag{5}$$

$$\begin{aligned} \text{PW-PadGen} \, (d_{v1} \, d_{v2} \, d_{v3} \, d_{v4}, R_t) &= a_{dt1} \, a_{dt1+16} \, a_{dt2} \, a_{dt2+16} \| a_{dm1} \, a_{dm1+16} \, a_{dm2} \, a_{dm2+16} \\ &\quad \| a_{dt3} \, a_{dt3+16} \, a_{dm3} \, a_{dm3+16} \| a_{dt4} \, a_{dt4+16} \, a_{dm4} \, a_{dm4+16} \text{ (Base 2)}, \\ &= k_{v1} \, k_{v2} \, k_{v3} \, k_{v4} \text{ (Base 16)}, \\ &= \text{PAD}. \end{aligned} \tag{6}$$

The Msg and password PW are XOR with the key to generate the cover coding message CCMsgx and cover coding password CCPWx. Through the XOR pad operation, the message and password can then recover their original form. The operation can be described as follows:

$$CCMsgx = PAD \oplus Msg, \tag{7}$$

$$CCPWx = PAD \oplus PW, \tag{8}$$

$$Msg = PAD \oplus CCMsgx, \tag{9}$$

$$PW = PAD \oplus CCPWx. \tag{10}$$

The above-discussed protocols have their advantages in terms of security; however, they have not been realized in wireless medical sensor networks for practical applications.

### 3.2. Definition of the Proposed Medication Safety WSN Protocol

As shown in Figures 1–3, the proposed protocol considers a multi-sensor node or gateway node environment as the network model where the participants in this model would be the users, with the hospital information system (HIS) and the pharmacy as the involved entities. We divided our scheme into three phases: registration phase, login/authentication phase, and password change phase. During this process, a genuine user produces their registration parameters over the network to the registration center (HIS), which is considered to be trusted. Successful registration of the user happens only after verification of the produced details. To access the information, the user produces their login credentials into the login/authentication phase. Then, the login request is received to verify the messages. In addition, a registered user executes the password change phase to update/modify their current password. In the beginning, the user sends a request to the HIS, two pairs of random numbers are generated, and these numbers are stored in the HIS. The details of the operation for each phase are explained below. Table 1 defines the nomenclature for the mutual authentication protocol.

**Table 1.** Customized authentication WSN symbol definition table.

| Symbol | Description |
|---|---|
| UserID | User identification |
| PWD | User password |
| RN | Random number |
| $Rt_x$, $Rm_x$ | Random number |
| Padgen () | Coding function |
| In_CCPW$_x$ | Input cover coding password |
| In_CCMsg$_x$ | Input cover coding message |
| Out_CCPW$_x$ | Output cover coding password |
| Out _CCMsg$_x$ | Output cover coding message |
| Req | Request |
| PAD$_x$ | Encryption pad |
| CRC | Cyclic redundancy code |
| $\oplus$ | XOR |
| $\parallel$ | Concatenation |
| NPWD | New password |

The first section is the registration phase, which includes 10 steps, as shown in Figure 1.

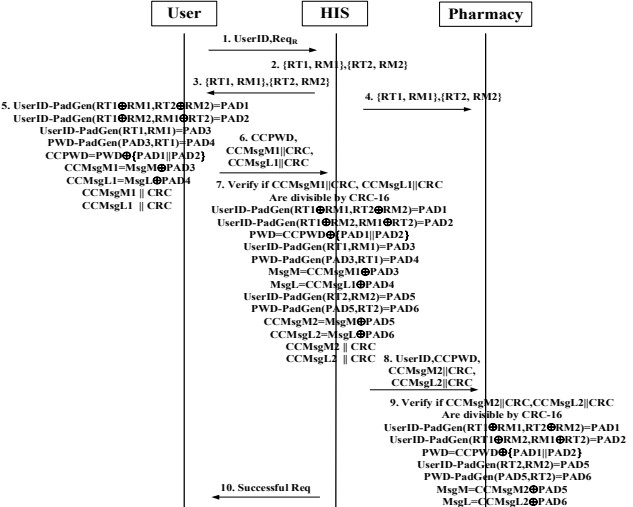

**Figure 1.** Registration phase.

The details of the operation are as follows:

Step 1: UserID and ReqR are sent to HIS from User, generating two sets of random numbers {RT1, RM1}, {RT2, RM2}.

Step 2: {RT1, RM1}, {RT2, RM2} random numbers are stored in HIS.

Step 3: Two sets of random numbers {RT1, RM1}, {RT2, RM2} are sent to User party.

Step 4: Same two sets of random numbers are sent to the Pharmacy party.

Step 5: PAD1, PAD2, PAD3, and PAD4 are obtained by selecting UserID or PWD with the specific random numbers for Padgen function operation. Then, CCMsgx is computed, followed by CRC encoding.

Step 6: CCPWD, CCMsgM1 ‖ CRC and CCMsgL1 ‖ CRC are sent to HIS.

Step 7: PAD1 to PAD6 are produced by selecting UserID or PWD with the specific random numbers for Padgen function operation. Afterward, through XOR operation, MsgM, MsgL, PWD, Out_ CCMsgM ‖ CRC, and out_CCMsgL ‖ CRC are obtained.

Step 8: UserID, CCPWD, CCMsgM2 ‖ CRC, and CCMsgL2 ‖ CRC are sent to Pharmacy party.

Step 9: PAD1, PAD2, PAD5, and PAD6 are obtained by selecting UserID or PWD with the specific random numbers for Padgen function operation. Then PWD, MsgM, and MsgL are computed through XOR operation.

Step 10: At last, a successful Req signal is sent to HIS, indicating completion of the phase.

The second section is the login/authentication phase, which includes 11 steps, as shown in Figure 2.

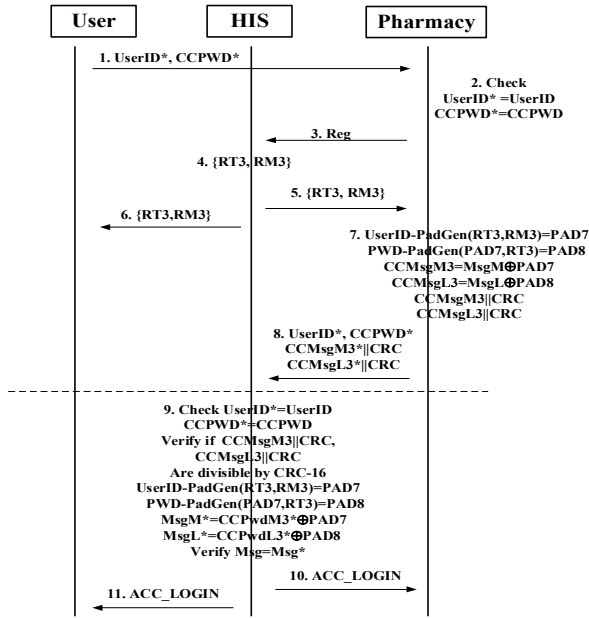

**Figure 2.** Login/authentication phase.

The details of the operation are as follows:

Step 1: UserID* and PWD* are entered by the User and sent to the Pharmacy party.

Step 2: Pharmacy identifies UserID* and PWD* for matching saved data.

Step 3: Reg is sent to HIS from Pharmacy.

Step 4: HIS generates a set of random numbers {RT3, RM3}.

Step 5: A set of random numbers {RT3, RM3} is sent to Pharmacy.

Step 6: A set of random numbers {RT3, RM3} is sent to the User.

Step 7: PAD7 and PAD8 are produced by selecting UserID or PWD with the specific set of random numbers for Padgen function operation. Afterward, through XOR operation, CCMsgM3 ‖ CRC and CCMsgL3 ‖ CRC are obtained.

Step 8: UserID, CCPWD, CCMsgM3 ‖ CRC, and CCMsgL3 ‖ CRC are sent to HIS party from Pharmacy.

Step 9: HIS checks UserID and CCPWD matching and verifies CRC code. PAD7 and PAD8 are produced by selecting UserID or PWD with the specific set of random numbers for Padgen function operation. Then, PWD, MsgM, and MsgL are computed through XOR operation.

Steps 10 and 11: At last, the ACC_LOGIN signal is sent to User and Pharmacy end for completion of the phase.

The third section is the password change phase, which includes six steps, as shown in Figure 3.

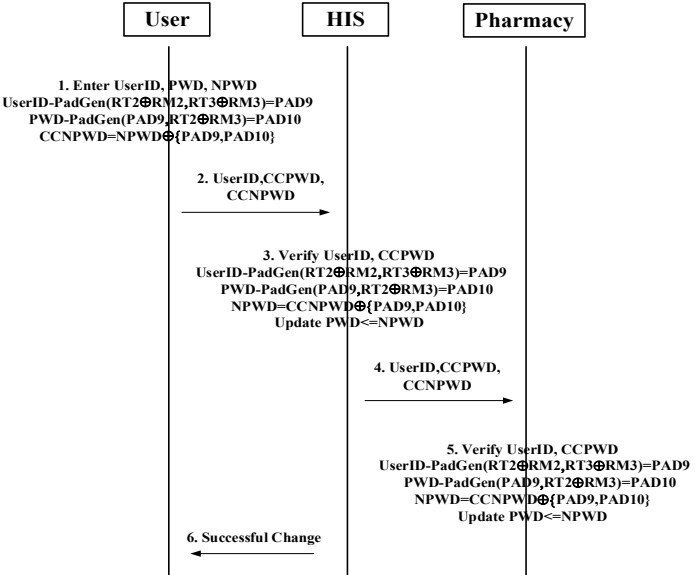

**Figure 3.** Password change phase.

The details of the operation are as follows:

Step 1: UserID, PWD, and NPWD begin calculation with UserID or PWD selected along with specific random numbers given to Padgen function for obtaining PAD9 and PAD10. Then, NPWD is encrypted by XOR operation, yielding CCNPWD.

Step 2: UserID, CCPWD, and CCNPWD are sent to HIS from PDA.

Step 3: HIS verifies the received UserID and CCPWD for matching. Afterward, calculation is initiated with UserID or PWD selected along with specific random numbers given to Padgen function for obtaining PAD9 and PAD10. Hence, NPWD is gained through XOR operation between the PADs and CCNPWD, and then it is stored and updated.

Step 4: UserID, CCPWD, and CCNPWD are sent to Pharmacy from HIS

Step 5: Similarly, the Pharmacy will verify the received UserID and CCPWD for matching. Afterward, calculation is initiated with UserID or PWD selected along with specific random numbers given to Padgen function for obtaining PAD9 and PAD10. Hence, NPWD is gained through XOR operation between the PADs and CCNPWD, and then it is stored and updated.

Step 6: Upon completion of updating NPWD, a signal successful change is sent to PDA.

## 4. Design and Implementation Results

### 4.1. XOR Method Medication Safety System Architecture

The block diagram of the medication safety WSN mutual authentication protocol (Medication Safety System) scheme with the XOR method is shown in Figure 4.

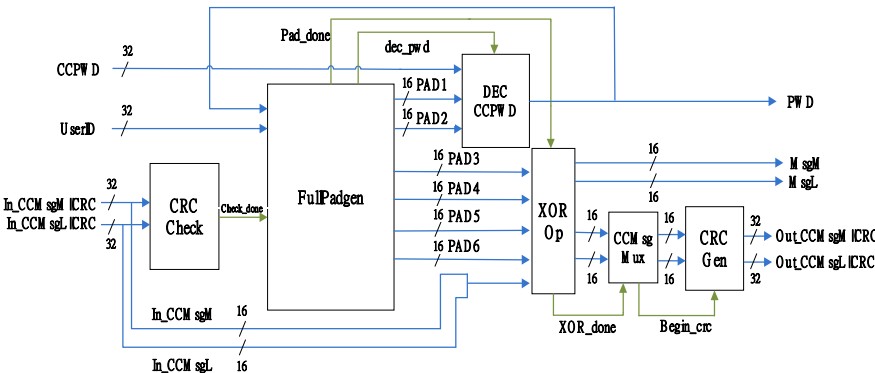

**Figure 4.** Block diagram of Medication Safety System (XOR method) scheme.

In Figure 4, the block diagram depicts the Medication Safety System (XOR method) scheme to verify the correctness of the received input cover-coded messages and passwords. The main function blocks include employing the cyclic redundancy check code method, generating the PADs required for decoding the received cover-coded password CCPWD, and inputting cover-coded messages. It also encodes the messages that are to be transmitted. The specific steps are as follows:

Step 1: Initially, the user's identification UserID, cover-coded password CCPWD, and a cover-coded message of the most and the least bits are concatenated with their cyclic redundancy check code. The received data inputs of medication safety system are In_CCMsgM ‖ CRC and In_CCMsgL ‖ CRC. Then, the CRC_Check module calculates and verifies whether both the input most-bit cover-coded message In_CCMsgM and the input least-bit cover-coded message In_CCMsgL are correct using their corresponding CRC code. A signal named Check_done of CRC_Check module is outputted, which resembles true (bit value of 0, as the remainder, is all zero) or false (bit value of 1, as nonzero reminders).

Step 2: Upon receiving input enable signal Check_done as a bit value of 0, FullPadgen module (0-active) can produce PAD1, PAD2, PAD3, PAD4, PAD5, and PAD6 in sequence under six control states as described later on in Figure 4. Meanwhile, when the FullPadgen module produces PAD2 on state {001}, an enable signal Dec_pwd is also sent to the DEC_CCPWD module. When enabled, the DEC_CCPWD module is used to operate decoding on CCPWD as shown below ( Equation (11)). The password PWD gained is fed back to the FullPadgen module for XOR-Padgen calculation of its 4 and 6 states.

$$PWD = CCPWD \oplus \{PAD1, PAD2\}. \tag{11}$$

Step 3: Once the FullPadgen module is completed with all six PADs produced, a Pad_done-enabling signal is given to the XOR_Op module, which begins to perform the XOR operation for decoding inputs In_CCMsgM and In_CCMsgL. After decoding, Msg (Equations (12) and (13)) is obtained, and then an internal signal is given followed by encoding to produce Out_CCMsgM and Out_CCMsgL (Equations (14) and (15)).

$$MsgM = In\_CCMsgM \oplus PAD3. \tag{12}$$

$$MsgL = In\_CCMsgL \oplus PAD4. \tag{13}$$

$$Out\_CCMsgM = MsgM \oplus PAD5. \tag{14}$$

$$Out\_CCMsgL = MsgL \oplus PAD6. \tag{15}$$

Step 4: CCMsg_Mux module collects Out_CCMsgM and Out_CCMsgL, and then gives the Begin_crc signal to CRC_Gen module, which allows it to process cyclic redundancy check code encoding for Out_CCMsgM and Out_CCMsgL. At last, Out_CCMsgM ‖ CRC and Out_CCMsgL ‖ CRC are outputted.

### 4.2. Verilog Simulation Results

The proposed Medication Safety WSN authentication protocol (XOR method) was implemented in a hardware description language (Verilog) on the Altera Quartus II platform, as shown in Figure 5.

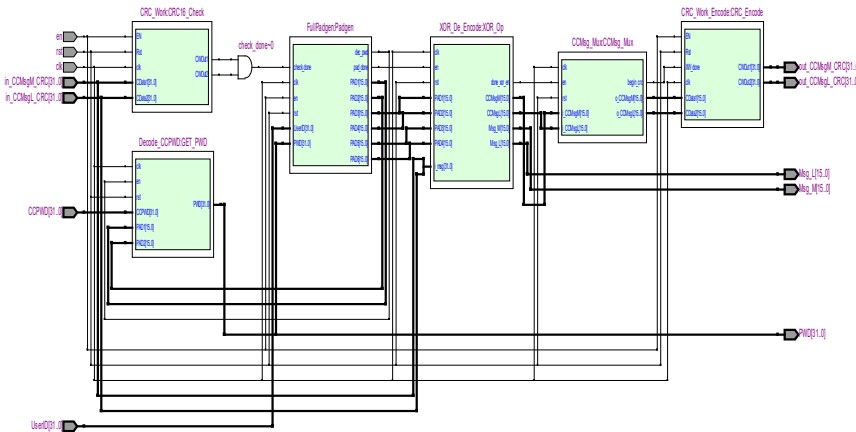

**Figure 5.** Hardware chart of Medication Safty WSN authentication protocol (XOR method).

The parameters of the design are reported in Table 2.

**Table 2.** The parameters of Medication Safty WSN authentication protocol hardware (XOR method).

| Family | Quartus II 8.0 Cyclone II |
|---|---|
| Device | EP2C70F896C6 |
| Logic elements | 1536/68,416 |
| Registers | 705 |
| PLLs | 0/4 |
| M4Ks | 0/250 |
| Total memory bits (Total RAM block bits) | 0/1,152,000 |
| Logic array blocks (Block interconnects) | 2063/197,592 |
| I/O pins | 259/622 |
| Clock pins | 1/8 |
| Maximum fan-out | 705 |
| Total fan-out | 8410 |
| Average fan-out | 2.91 |
| Embedded multipliers | 0/300 |
| FPGA clock source (MHz) | 50 |
| Fmax (MHz) | 174.40 |
| Clock period (ns) | 6.281 |
| Longest Path Delay (ns) | 5.512 |
| Worst Case Setup time ($t_{su}$) (ns) | 7.214 |
| Worst Case Hold time ($t_h$) (ns) | 0.496 |
| Total Power (mW) | 246.07 |
| Dynamic Power (mW) | 20.79 |
| Static Power (mW) | 155.13 |
| I/O Power (mW) | 70.15 |

Simulations of the proposed design were conducted in the Altera Quartus II design environment and implemented in an Altera Cyclone II EP2C70F896C6 FPGA on an Altera DE 2 board. The simulation results of the XOR-Padgen function as described in Section 4.1 are shown in Figure 6.

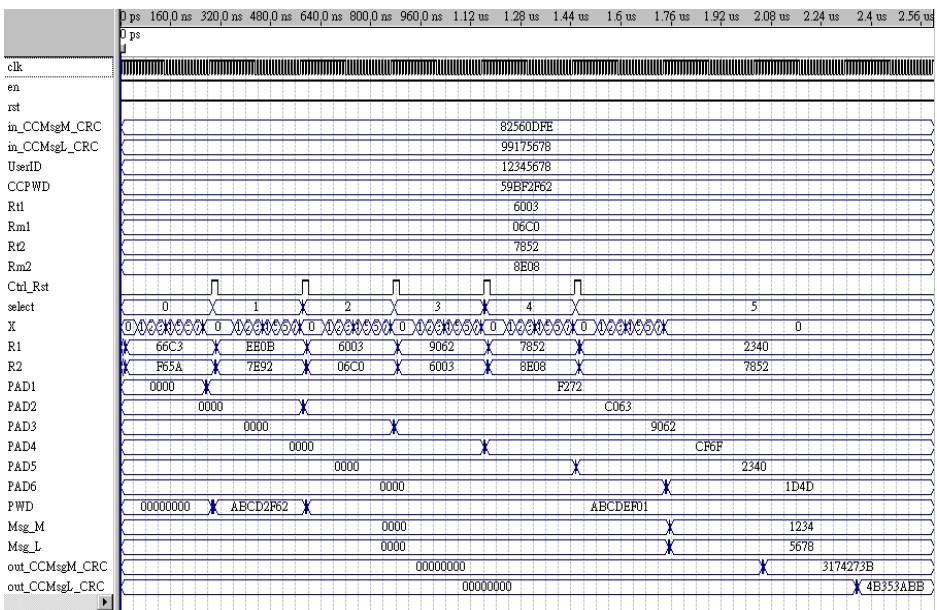

**Figure 6.** Simulation Results of the XOR-Padgen function.

The received user's identification and cover-code password are assumed to be UserID = 12345678 and CCPWD = 59BF2F62. The two input numbers R1 and R2 for Padgen function are produced by XOR operation between two random numbers; for example, when R1 = Rt1 ⊕ Rm1 = 66C3 and R2 = Rt2 ⊕ Rm2 = F65A, the calculated result PAD1 = F272 at select = 0. When R1 = Rt1 ⊕ Rm2 = EE0B and R2 = Rm1 ◌̇ Rt2 = 7E92, the calculated result PAD2 = C063 at select = 1. When R1 = Rt1 = 6003 and R2 = Rm1 = 06C0, the calculated result PAD3 = 9062 at select = 1. When R1 = PAD3 = 9062 and R2 = Rm1 = 6003, the calculated result PAD4 = CF6F at select = 2. When R1 = Rt2 = 7852 and R2 = Rm2 = 8E08, the calculated result PAD5 = 2340 at select = 3. When R1 = PAD5 = 2340 and R2 = Rt2 = 7852, the calculated result PAD6 = 1D4D at select = 5. PWD = ABCDEF01 is gained by decoding CCPWD which is computation of CCPWD ⊕ {PAD1, PAD2}. MsgM = 1234 is obtained by calculating in_CCMsgM = 8256 ⊕ PAD3, and MsgL = 5678 is obtained by calculating in_CCMsgL = 9917 ⊕ PAD4. out_CCMsgM = 3174 is gained by operating on MsgM ⊕ PAD5, and out_CCMsgL = 4B35 is gained by operating on MsgL ⊕ PAD6. Once out_CCMsgM and out_CCMsgL are produced, an internal signal is given to begin its CRC encoding, thus generating CCMsgM ‖ CRC (=273B) and CCMsgL ‖ CRC (=3ABB).

### 4.3. FPGA Hardware Verification of the XOR-Padgen Algorithm

Simulations of the proposed design were conducted in the Altera Quartus II design environment. The verified Verilog code was then downloaded on an Altera Cyclone II EP2C70F896C6 FPGA running with a 50 MHz clock in the Altera DE2 board to verify the hardware. The output waveforms from the FPGA are displayed using the HP 16702A logic analysis system for real-time verification. Figure 7 shows eight seven-segment LEDs representing the input number for the Padgen function. Subsequently, the decoded message is exhibited on the first line of the LCD, while the most- and least-bit output cover-coded messages are produced and shown on the second line of the LCD. In addition, Figure 7 shows that the Altera DE2-70 board (left-hand side of Figure 7) switches variation from Sw = 2'b00 to Sw = 2'b11. The four states command the data to be sent, thus obtaining pin measurements on the HP 16702A logic analysis system (right-hand side of Figure 7); its outputs in sequence are UserID = 12345678, CCPWD = 59BF2F62, Out_CCMsgM ‖ CRC (Out_CCMsgM) = 3174273B (hex), and Out_CCMsgL ‖ CRC (Out_CCMsgL) = 4B353ABB (hex) (Figure 7a–d).

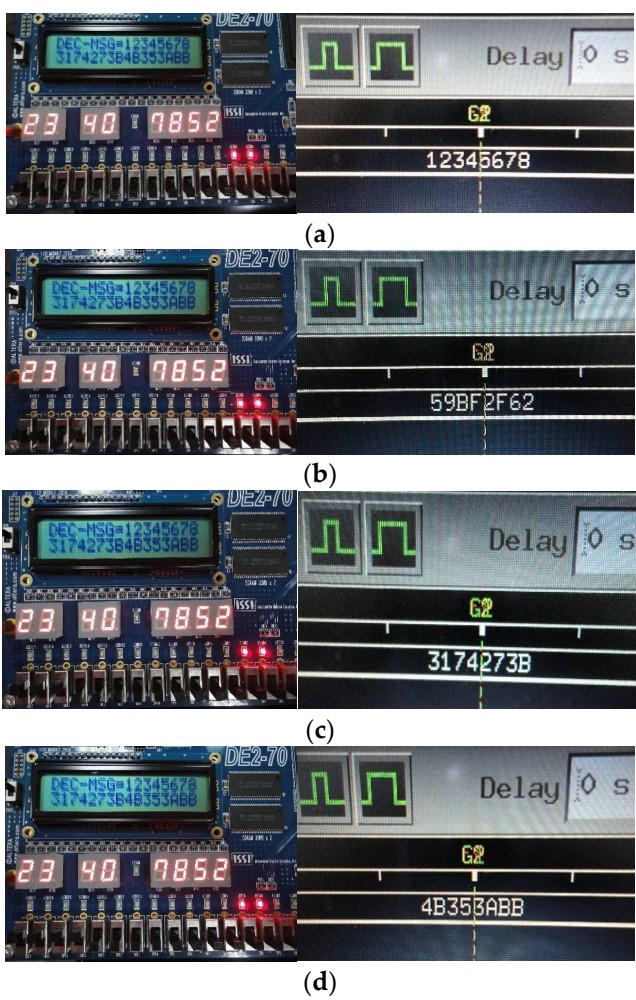

**Figure 7.** FPGA Hardware verification of the functionality of the proposed protocol. (**a**) When Sw = 2′b00, outputs UserID; (**b**) When Sw = 2′b01, outputs CCPWD; (**c**) When Sw = 2′b10, outputs CCMsgM ‖ CRC; (**d**) When Sw = 2′b11, outputs CCMsgL ‖ CRC.

## 5. Conclusions

Sensor networks are used in several domains that handle sensitive information. In this paper, the main objective in the security design of the described scenario was data authentication, whereby a WMSN should be able to verify whether sensor data originated from legitimate sensors. A medication safety mutual authentication protocol with the Padgen algorithm was proposed to improve the drawbacks of the authentication scheme in WSN. Through the Padgen algorithm, the message can be cover-coded and security can be enhanced during wireless data transmission. FPGA hardware verification of the proposed architecture was also demonstrated; the proposed architecture has a more efficient area and lower power than the traditional authentication scheme. For the integration of the proposed protocol in an electronic system, the interfaces of the integrated system components need to be further studied to realize wireless data transmission across system integration.

**Author Contributions:** Conceptualization, Y.-J.H.; methodology, W.-C.L. and Y.-J.H.; software, P.-K.H. and C.-L.P.; validation, W.-C.L. and Y.-J.H.; formal analysis, P.-K.H.; investigation, C.-L.P. and Y.-J.H.; data curation, P.-K.H. and W.-C.L.; writing—original draft preparation, W.-C.L. and Y.-J.H.; writing—review and editing, Y.-J.H.; visualization, P.-K.H.; supervision, Y.-J.H.; project administration, Y.-J.H.; funding acquisition, Y.-J.H. All authors read and agreed to the published version of the manuscript.

**Funding:** This work was supported in part by the EDA healthcare under grant EDDHM109002, ISU-110-IUC-10 and the Ministry of Science and Technology of Taiwan under grant MOST 110-2221-E-214-025.

**Data Availability Statement:** Data is contained within the article.

**Conflicts of Interest:** The authors declare no conflict of interest.

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
