# Peer review of "FPGA Implementation of Mutual Authentication Protocol for Medication Security System"

_jlpea, doi:10.3390/jlpea11040048_

Round 1
Reviewer 1 Report
- In keywords, I suggest using a keyword indicating that the paper is concerned with a problem related to the medical sphere. For example: wireless medical sensor networks or another appropriate word could be added as a keyword.
- Line 35 - This sentence seems to be written in a smaller font than the rest of the paper. " The integration of WSN applications and low-power sensing nodes with the Internet has been accomplished with various approaches and strategies"
- Line 52 - It is stated: "If the wireless transmission of information exchange has not been protected properly, the transmission of data messages will be very vulnerable when an adversary carries out attacking, eavesdropping, stealing, intercepting, and tampering on communication between sensor nodes, which could lead to incorrect or altered messages delivery, in result causes a severe medical casualty". I recommend citing some papers (if there are any) where it is analyzed how serious this problem is.
- Could you discuss more precisely how sensors for medicine purposes differ from sensors applied in other areas (e.g., forest fire detection, flood detection, pollution monitoring, etc.)?
- Line 107 - 115 - It seems that the space between the lines is bigger than in the rest of the paper. This should be unified.
- Line 115, Line 120 - WSMN is used instead of WMSN
- Line 117 - 147 - The text in this part is not aligned.
- Line 122 - space is missing in "are32-bit message data"
- Line 133 - different font in the label of (2.5). "5" has a different font.
- Line 230 - space is missing in " Figure4"
- Line 308 - probably an incorrect operator in Rt1♁Rm1 = 66C3. This should be checked by the authors.
- In Conclusion, I recommend focusing your attention primarily on your contribution. The general theory should be reduced from this part as it is discussed in the paper. Thus, the text in Line 346 - 354 should be shortened.
Reviewer 2 Report
The authors present an implementation of a field-programmable gate array to ensure security, privacy, and reliability of medication information over a WSN. Parts of the paper are really well-written, but other parts seem rushed or incomplete, so I recommend a major revision.
On lines 37-40, the authors discuss what they perceive as "merits" of WSNs. This may be a language issue, but the items listed are not really merits and are instead challenges. For example, one common challenge of WSNs is high power consumption but the authors list "low power consumption" as a merit. There is a significant literature about reducing power consumption to extend the lifetime of WSNs because the opposite of what the authors claim. Similarly, the authors list "distributed sensors". This may be more of a perception of the problem. The distributed nature of WSNs often requires novel (perhaps application dependent) approaches to the data processing, so that can actually be viewed as a challenge as well, rather than a merit. So this claim of the authors is confusing and should be explained further to clarify their meaning or should be corrected.
In the context of privacy, the authors might consider referencing legal requirements of privacy of certain data, e.g., HIPAA in the United States or PDPA in Taiwan, and consider if their approach has any potential issues or concerns with compliance.
The authors should carefully describe all abbreviations. For example, "PDA". This likely may be a personal digital assistant, but who is the user exactly (the patient), who has ownership of the device, and what are the requirements of the PDA.
There is no discussion of the registration, authentication and password change phases in section 3.2. A list of steps is insufficient to include in the text. Describe the purpose of each step similar to how the steps of the XOR scheme is described in section 4.1. It may be easier to describe the protocol in narrative form and include parenthetical references to the steps, e.g. "... the user sends a request to the HIS and 2 pairs of random numbers are generated (Step 1). These numbers are stored in the HIS...".
There are a number of issues that indicate the paper is still in an early draft state, possibly with content that was copied from an earlier submission or work. Several examples of this are described below:
1) On lines 32 and 34, the authors use "(DAR 1978)" and "[3]" in the same sentence to reference the same item in the reference list. This is mixing LNI and IEEE citation styles.
2) Starting in section 3 on line 107, the line spacing suddenly changes and then reverts back at line 117. However, at line 117 the text is no longer full justified and is only left justified.
3) The font of "3.1" in the subsection heading of line 116 is different from the font of "4.1" on line 228.
4) There are a few places missing spaces or with extra spaces. For example, on line 122, there is no space between the "e" and "3" in "are32-bit", and there is an extra space between "as" and the following colon.
5) The equation number positioning is inconsistent in lines 133 and 137.
6) In line 153, several figures are referenced as "figures 3.1 to 3.4". The figures are actually labeled 1 to 4.
7) In section 3.2, there are several sentences that are awkwardly phrased. For example, on line 150 "to have lower the computational cost" and line 153 "To propose improved the computational cost". This creates the appearance that the text was partially and incompletely edited.
8) Lines 204-205 are repeated in lines 206-207.
Reviewer 3 Report
Specific comments and recommendations:
1) Sections 1 and 2 of this work are too general written, with no technical analysis of the problem. How exactly is the goal and objectives of the research defined?
2) Why exactly did the authors choose the implementation with FPGA integrated circuits - the authors have to point out some advantages and disadvantages of this solution;
3) Are there other methods in the literature to study those problems? It is good if such methodologies exist to cite and comment on their advantages and disadvantages;
4) How is the proposed protocol to ensure the safety of medicines - what standards and norms are observed in this case?
5) The authors have to show how to integrate the proposed protocol in an electronic system for wireless data transmission, as well as to present experimental results from the real operation of the system.
Reviewer 4 Report
This paper presents a user authentication scheme for healthcare information systems based on a wireless
sensor network.
The proposed scheme exploits the Padgen algorithm as the basis for mutual authentication.
The resulting medication safety WSN authentication protocol is divided into three phases:
registration, login/authentication, and password change which are described in detail in section 3.
Design and FPGA implementation are reported in section IV.
Validation is based on both simulations and measurements on an FPGA evaluation board.
Resource usage is reported in Table 2 highlighting the compactness of the design.
Some more details about the implementation of different modules would further improve the manuscript.
The paper is enough clear and well written even if english should be a little bit revised.
Round 2
Reviewer 1 Report
The paper can be accepted in its current form in my opinion.
Author Response
Thanks for the reviewer’s suggestion, we have carefully read and spell-checked the updated manuscript.
Reviewer 2 Report
The edits are generally sufficient in my opinion. The authors should review carefully again as there are some redundancy or typo errors in the edits that should be corrected. In lines 39-40, there are ellipses (“…”) followed by an “etc.”. They essentially mean the same thing. Don’t use the ellipses. In line 83, “Health Insurance Portability and Accountability (HIPAA) Act” should be “Health Insurance Portability and Accountability Act (HIPAA)” and “Personal Data Protection (PDPA) Act” should be “Personal Data Protection Act (PDPA)”.
Author Response
We would like to thank the reviewers for their thoughtful comments and efforts towards improving our manuscript.
We apologize for these typo errors which have been corrected in the updated manuscript.

Reviewer 3 Report
In the present version of the manuscript in the majority, the recommendations are implemented. In my opinion, the manuscript can be accepted after minor revision for publication. In my opinion, the authors have to consider the possible integration of the proposed protocol in an electronic system for wireless data transmission, as well as to present experimental results from the real operation of the system. The presentation of experimental results best confirms the significance and effectiveness of innovative work. Also, the quality of Fig. 5 and Fig. 6 needs to be significantly improved, in this form the data in the figures are illegible.
Author Response
We thank the reviewer for calling attention to this. We add the following sentences in the conclusions to address the concern of the reviewer.
- For the integration of the proposed protocol in an electronic system, the interface among the integrated system components needs to be further studied to realize the wireless data transmission cross the system integration.
- 5 and Fig. 6 are improved.
